civil engineering/computational mechanics/
artificial intelligence

aerodynamic coefficients, iced quad bundle
conductor, prediction model, galloping,
machine learning

**Author for correspondence:**
Bo Yan
e-mail: boyan@cqu.edu.cn

# Prediction model for aerodynamic coefficients of iced quad bundle conductors based on machine learning method

Zheyue Mou[1], Bo Yan[1], Hanxu Yang[1], Daoda Cai[1] and Guizao Huang[2]

[1]College of Aerospace Engineering, Chongqing University, Chongqing 400044, People's Republic of China
[2]School of Electrical Engineering, Southwest Jiaotong University, Chengdu 611756, People's Republic of China

ZM, 0000-0002-4947-9513

The lift, drag and torsional moment coefficients, versus wind attack angle of iced quad bundle conductors in the cases of different conductor structure, ice and wind parameters are numerically simulated and investigated. With the Latin hypercube sampling and numerical simulation, sampling points are designed and datasets are created. Set the number of subconductors, wind attack angle, bundle spacing, ice accretion angle, ice thickness, wind velocity and diameter of the conductor as the input variables, a prediction model for the lift, drag and moment coefficients of iced quad bundle conductors is created, trained and tested based on the dataset and extra-trees algorithm. The final integrated prediction model is further validated by applying the aerodynamic coefficients from the prediction model and numerical simulation, respectively, to analyse the galloping features. The developed efficient prediction model for the aerodynamic coefficients of iced quad bundle conductors plays an important role in the quick investigation, prediction and early warning of galloping.

## 1. Introduction

Galloping accidents of iced transmission lines take place frequently and may give rise to flashover, breakage of conductors, collapse of towers and disruption of power supplies. Quick investigation and prediction of galloping features of transmission lines are very important for the development of anti-galloping technique and

early warning system. The aerodynamic forces, namely the lift, drag and torsional moment, acting on the iced conductors with non-circular section in wind field play key role in the inducing of galloping of transmission lines [1]. The aerodynamic coefficients of iced conductors depend on many parameters such as the diameter of sub-conductor, spacing between sub-conductors, ice shape and dimension, ice accretion angle, wind velocity and attack angle. Because these parameters may change in a wide range, the determination of the aerodynamic coefficients with wind tunnel tests and numerical simulation is too expensive and time-consuming. Therefore, the creation of a prediction model to quickly determine the aerodynamic coefficients of iced conductors under different parameters is urgent for the investigation and prediction of galloping features as well as the development of anti-galloping technique and early warning system.

Wind tunnel tests were usually used to measure the aerodynamic coefficients of iced conductors by many authors. Nigol & Buchan [2] and Chabart & Lilien [3] investigated the aerodynamic characteristics of iced single conductors with wind tunnel test. Li [4] studied the aerodynamic characteristics of two types of iced conductors by means of quasi-steady and dynamic wind tunnel tests and concluded that the quasi-steady aerodynamic forces can be used to investigate the galloping phenomenon. Lou *et al.* [5] carried out wind tunnel tests to obtain the aerodynamic coefficients of single and bundle conductors with different thickness of ice and identified the ranges of wind attack angles sensitive to galloping. The group of the authors of this paper took into account the wake influence of the windward sub-conductors to the aerodynamic behaviour of the leeward sub-conductors, and measured the aerodynamic coefficients of each iced sub-conductor of the quad bundle and eight bundle conductors with crescent-shaped and sector-shaped ice under various wind velocities. It is found that the aerodynamic coefficients of the sub-conductors are different from each other and the difference will affect the initiation and behaviour of galloping of iced bundle conductor lines [6,7]. Recently, the wind tunnel tests were used to measure the aerodynamic coefficients of conductors with different shaped ice by other authors [8–10].

On the other hand, the computational fluid dynamics method has been employed to analyse the air flow around conductors. Braun & Awruch [11] proposed a numerical model for the aerodynamic and aeroelastic analysis of bundle conductors. The group of the authors of this paper numerically determined the aerodynamic coefficients of a quad bundle conductor accreted with crescent-shaped ice [12] and those of iced twin bundle conductors varying with relative location between two sub-conductors [13] using a two-dimensional model by the Fluent software. Ishihara & Oka [14] computed the aerodynamic coefficients of single and quad bundle ice-accreted conductors using the large-eddy-simulation (LES) turbulence model and presented the correction coefficients for the leeward sub-conductors to reflect the wake influence. However, there are some deviations between the results by the numerical simulation and those by the wind tunnel tests. To arrive at enough accuracy of the numerical simulation, very fine mesh of the model is required, leading to lower efficiency and high computational cost.

Recently, the application of artificial intelligence in fluid dynamics has attracted much attention [15]. Based on the machine learning and deep learning methods, the prediction models for aerodynamic characteristics can be established to map the complicated nonlinear relations between the aerodynamic coefficients and the parameters of the flow field and structure. Linse & Stengel [16] combined the neural network with simulated flight data firstly to generate a model for the accurate identification of aerodynamic coefficients of a twin-jet aircraft. Chung & Alonso [17] constructed approximation models by the Cokriging method and optimized the aerodynamic design of a supersonic jet. Mackman & Allen [18] applied the radial basis function (RBF) interpolation and developed multi-criteria adaptive sampling method to create models for lift, drag and moment coefficients of an aerofoil across ranges of the Mach number, Reynolds number and wind attack angle. Han *et al.* [19] proposed a high-efficiency global optimization method by combining the multi-level hierarchical Kriging (MHK) prediction model with the expected improvement function and validated the method by benchmark aerodynamic shape optimization of two aerofoils with dozens of design variables. Based on the convolutional neural network (CNN), Chen *et al.* [20] presented a graphical model, in which the transformed aerofoil image was set as input, and the model can predict the pitch-moment, lift and drag coefficients with high accuracy.

It is known that the aerodynamic coefficients of iced bundle conductors depend on many parameters, including the conductor line structure, accreted ice and wind, and the relationships between the coefficients and these parameters are very complicated. In this paper, a prediction model for aerodynamic coefficients of iced quad bundle conductors will be created based on the aerodynamic data and machine learning method. The Latin hypercube sampling (LHS) method is used to design

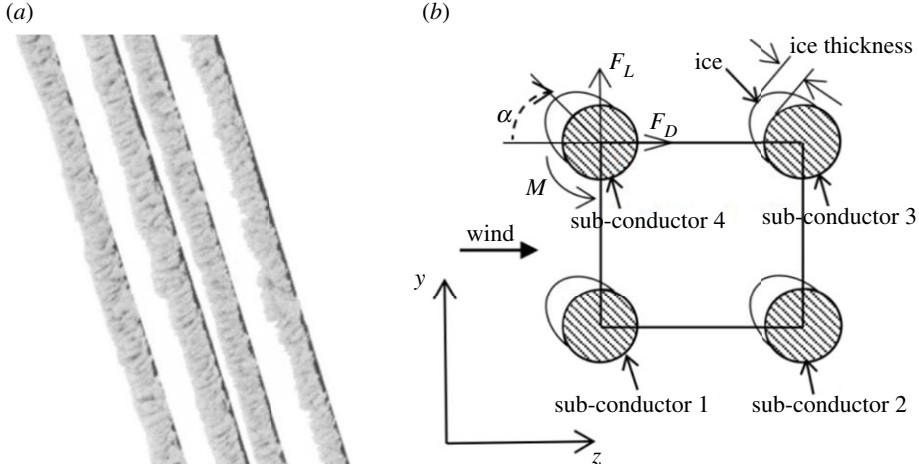

**Figure 1.** Quad bundle conductor with crescent-shaped ice. (a) Real quad bundle conductor with ice and (b) cross-section model of iced quad bundle conductor.

sampling points for numerical simulation of aerodynamic coefficients of iced quad bundle conductors, and the results are used to construct the datasets. Five variables, i.e. the bundle spacing, ice accretion angle, ice thickness, wind velocity and diameter of the conductor, are set as the input of the prediction model, and the curves of the lift, drag and torsional moment varying with wind attack angle as the output. The created model can predict the three aerodynamic coefficients of iced quad bundle conductors efficiently and quickly, and it can be used to investigate the galloping behaviour of iced transmission lines, and develop anti-galloping devices and early warning system for galloping.

# 2. Aerodynamic coefficients of iced quad bundle conductors

To create prediction model for aerodynamic coefficients of iced quad bundle conductors, the lift, drag and torsional moment coefficients of iced quad bundle conductors with specific parameters should be determined. The data obtained with wind tunnel tests are very limited, and thus the numerical simulation method for air flow around bundle conductor is used. The group of the authors of this paper investigated the aerodynamic characteristics of iced quad bundle conductors by means of the computational dynamics method [12], and the efficiency of the method was demonstrated with the wind tunnel test results.

## 2.1. Numerical simulation of air flow around iced conductor

The crescent-shaped ice, as shown in figure 1a, is frequently observed in the iced transmission lines with galloping taking place. To analyse its aerodynamic characteristics, a two-dimensional model as shown in figure 1b is created. Each iced sub-conductor is subjected to aerodynamic forces under wind action, including the lift $F_L$, drag $F_D$ and torsional moment $M$. The wind attack angle is defined as the angle between the wind direction and the centre line of the accreted ice while the ice accretion angle, i.e. the growth direction of the ice, is defined as the angle between the centre line of the accreted ice and the negative $z$-axis.

Recently, the group of the authors secondarily developed a simulation software for aerodynamic characteristics of iced conductors, which calls the commercial computational fluid dynamics software Fluent as solver to simulate the air flow around the iced conductor, as shown in figure 2. With the secondarily developed software, the geometrical models of iced conductors with any shaped ice can be constructed quickly and the curves of the three aerodynamic coefficients varying with wind attack angle can be output automatically.

A numerical simulation model of an iced quad bundle conductor created using the software is shown in figure 3a. The conductor type is $4 \times$ LGJ-400/50, and the diameter of each sub-conductor is 27.6 mm. The bundle spacing between two adjacent sub-conductors is 450 mm and the size of the square domain for the flow simulation is $6 \times 6$ m. The numerical models with ice thickness of 12 mm, 20 mm and 28 mm are investigated, respectively. The wind velocity is 12 m s$^{-1}$, and the wind attack angle ranges from 0° to 360° with an increment of 5°. The domain is discretized with quadrilateral elements as showed in figure 3b, and the finite volume method is used. In addition, the Spalart–Allmaras turbulent model is

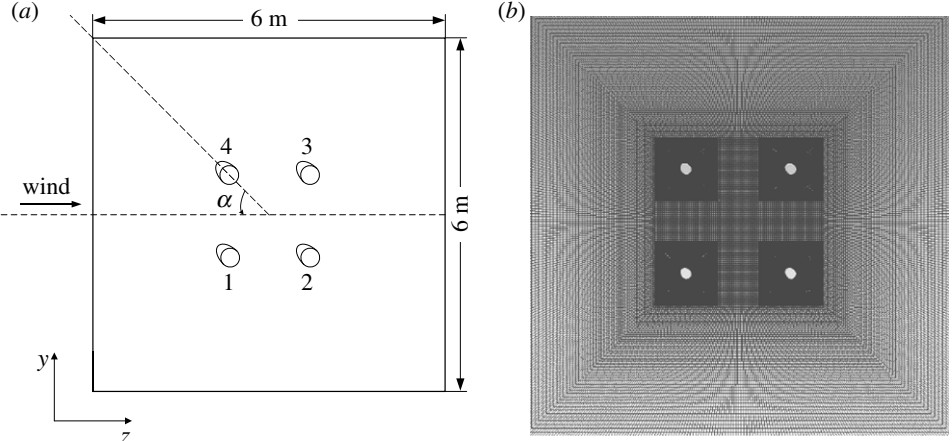

**Figure 2.** Developed simulation software for aerodynamic characteristics of iced conductors. (*a*) Conductor parameter setting; (*b*) wind parameter setting; (*c*) ice parameter setting and (*d*) computational parameter setting.

**Figure 3.** Numerical models of iced quad bundle conductor. (*a*) Simulation model and (*b*) mesh model.

adopted to describe the turbulence of aerodynamic wind flow, and the SIMPLE algorithm and three-order-precision QUICK scheme are adopted for the analysis of the flow field. The simulated velocity contours of the iced quad bundle conductor under different wind attack angles are shown in figure 4. It is noted that the average simulation time for obtaining the curves of the aerodynamic coefficients of an iced quad bundle conductor varying with wind attack angle is more than 30 h using computer ThinkCentre M8600t with Intel(R) Core i7-6700.

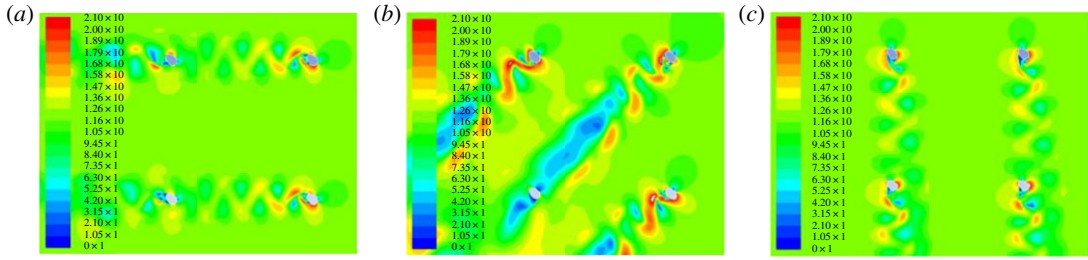

**Figure 4.** Velocity contours of the iced quad bundle conductor under different wind attack angles. (*a*) Wind attack angle: 225°; (*b*) wind attack angle: 270° and (*c*) wind attack angle: 315°.

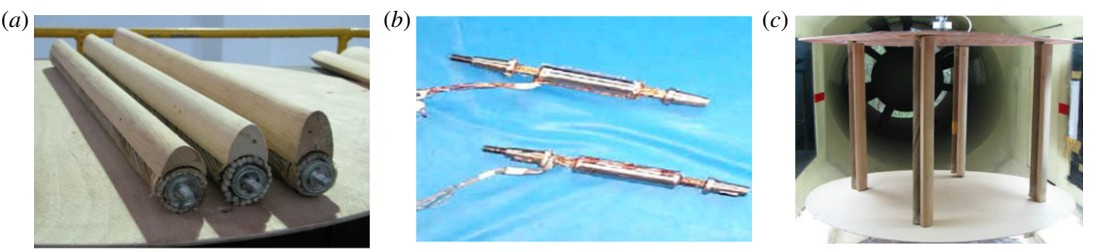

**Figure 5.** Wind tunnel tests for the iced quad bundle conductors. (*a*) Iced sub-conductor models; (*b*) strain balances and (*c*) test system.

## 2.2. Wind tunnel tests for aerodynamic coefficients

The research group carried out wind tunnel tests to measure the aerodynamic coefficients of iced quad bundle conductors with crescent-shaped ice, and the test system in the wind tunnel is shown in figure 5. The details of the wind tunnel tests can be found in the work of Hu *et al.* [6]. The artificial crescent-shaped ice is made of light wood while the sub-conductors are modelled by aluminium tubes with screwed plastic wires. The strain balances are installed inside the cavity of each sub-conductor model to measure the three aerodynamic forces, including lift, drag and torsional moment. The quad bundle conductor model with length of 700 mm is connected with two circular plates, which are used to keep the two-dimensional inflow. The aerodynamic forces of each sub-conductor under different wind attack angles are obtained by rotating the test model with an increment of 5°. It is noted that the ice and wind parameters are the same as in the numerical model as discussed in §2.1.

## 2.3. Comparison of numerical and test results

The aerodynamic coefficients, including the lift, drag and moment coefficients of an iced sub-conductor, can be defined, respectively, as the follows:

$$C_L = \frac{F_L}{(1/2)\rho_{\text{air}}V^2LD} \quad C_D = \frac{F_D}{(1/2)\rho_{\text{air}}V^2LD} \quad C_M = \frac{M}{(1/2)\rho_{\text{air}}V^2LD^2}, \tag{2.1}$$

where $F_L$, $F_D$ and $M$ are the lift, drag and torsional moment, respectively, $\rho_{\text{air}}$ is the density of the air, $V$ is the wind velocity, $L$ is the length of the sub-conductor, $D$ is the diameter of the bare sub-conductor.

Comparison of the aerodynamic coefficients of sub-conductor 1 of the iced quad bundle conductors with different ice thickness under wind velocity of 12 m s$^{-1}$ varying with wind attack angle is shown in figure 6. It can be seen that the changing laws of the lift, drag and moment coefficients determined by the numerical simulation and the wind tunnel tests are consistent. The differences between the aerodynamic coefficients obtained by the two methods may be induced by the simplification of stranded structure of conductor in the numerical model, and the stability of inflow wind field and measurement errors in the wind tunnel tests. However, the galloping of transmission lines depends on the changing laws of the aerodynamic coefficients, and the Den Hartog coefficient Dc and Nigol coefficient Nc [2], which are defined as

$$Dc = \frac{\partial C_L}{\partial \alpha} + C_D \tag{2.2}$$

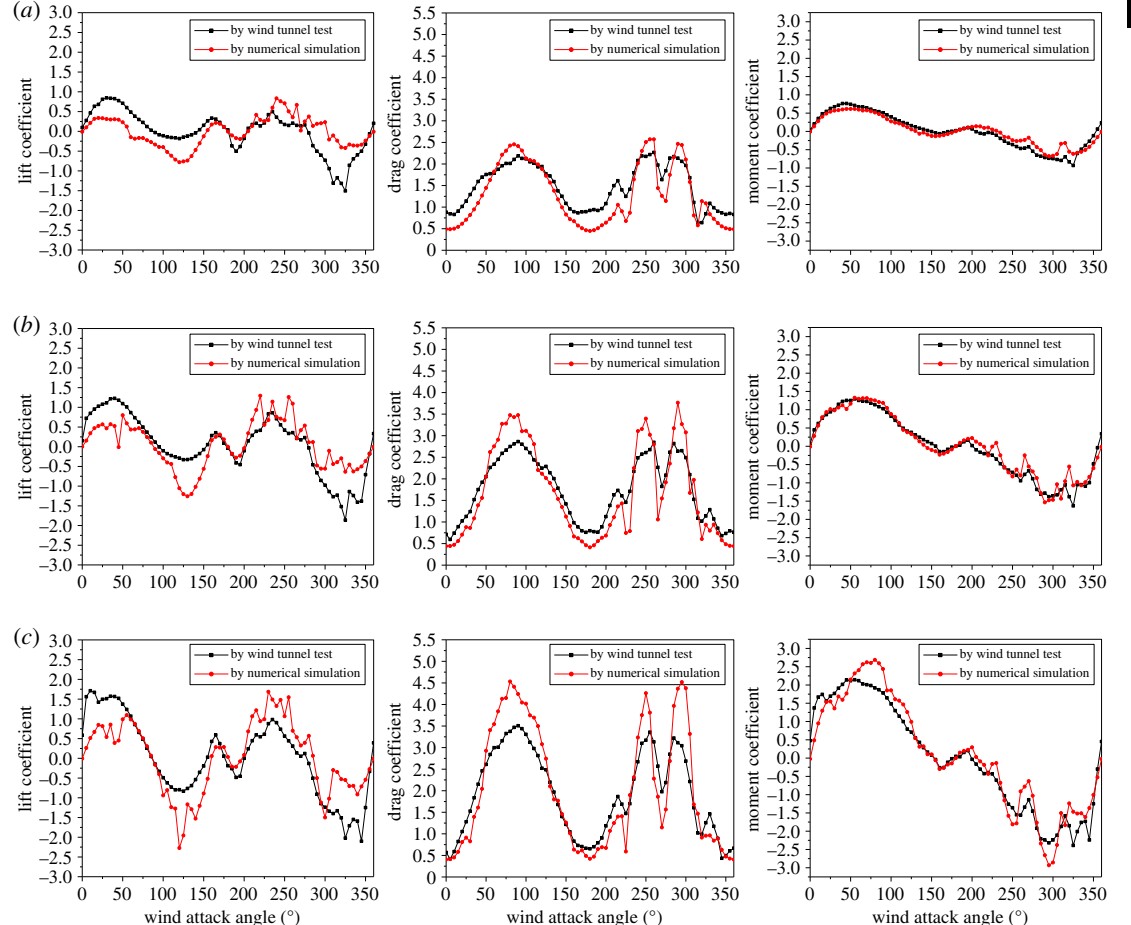

**Figure 6.** Comparison of aerodynamic coefficients of sub-conductor 1 varying with wind attack angle in iced quad bundle conductors with different ice thickness under wind velocity of 12 m s⁻¹. (*a*) Ice thickness: 12 mm; (*b*) ice thickness: 20 mm and (*c*) ice thickness: 28 mm.

and

$$\mathrm{Nc} = \frac{\partial C_M}{\partial \alpha}, \qquad (2.3)$$

where $\alpha$ is the wind attack angle. The galloping of iced transmission lines may be induced when $\mathrm{Dc} < 0$ (vertical galloping mechanism) or $\mathrm{Nc} < 0$ (torsional galloping mechanism).

The Den Hartog coefficients and the Nigol coefficients of the four sub-conductors with 12 mm thickness ice against the wind attack angle based on numerical simulation and wind tunnel test are compared in figure 7. It is seen that the curves of these two coefficients determined by the two methods are similar. The galloping of a transmission line with an initial wind attack angle of 45° was simulated based on the aerodynamic coefficients by the two methods in our previous works [12]. From the curves of the four sub-conductors, the Den Hartog coefficient of all or some sub-conductors are negative in the angle ranges of 160–190°, 265–280° and 310–350°, in which the vertical galloping may take place. Moreover, the Nigol coefficients of some sub-conductors in these attack angle ranges are negative. In order to prove the acceptance of aerodynamic coefficients by numerical simulation further, the dynamic responses of a line with crescent-shaped ice under wind load are simulated. The finite-element model of the quad bundle conductor line with span length of 200 m is shown in figure 8. The conductor type is LGJ-400/50, the initial conductor tension is 27.09 kN and the bundle spacing is 450 mm. The ice thickness is 12 mm and the wind velocity is 12 m s⁻¹. The initial wind attack angles are changed in the ranges of 160–190°, 265–280° and 310–350° with an increment of 5°. The aerodynamic coefficients obtained by numerical simulation and wind tunnel test are, respectively, used.

The simulated results based on the aerodynamic coefficients by the two methods both reflect that the vertical galloping takes place when the initial wind attack angle locates in the range of 170–185°, and no

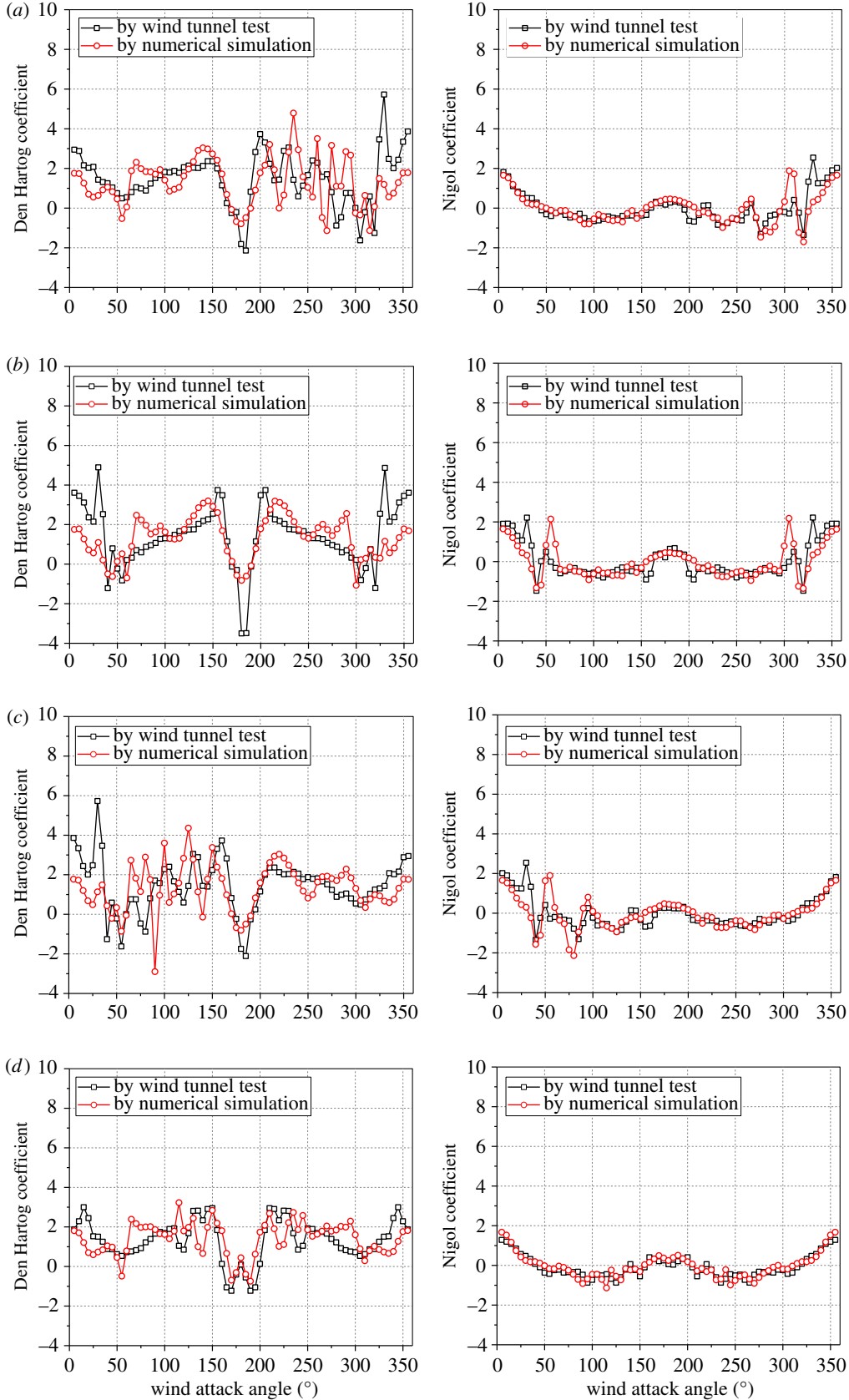

**Figure 7.** Den Hartog coefficients and Nigol coefficients of the iced quad bundle conductors with 12 mm thickness ice under wind velocity of 12 m s$^{-1}$. (*a*) Sub-conductor 1; (*b*) sub-conductor 2; (*c*) sub-conductor 3 and (*d*) sub-conductor 4.

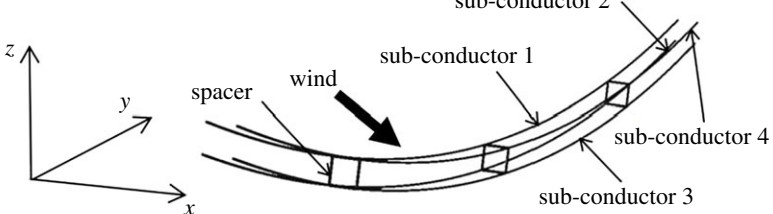

**Figure 8.** Finite-element model of a typical iced quad bundle conductor line.

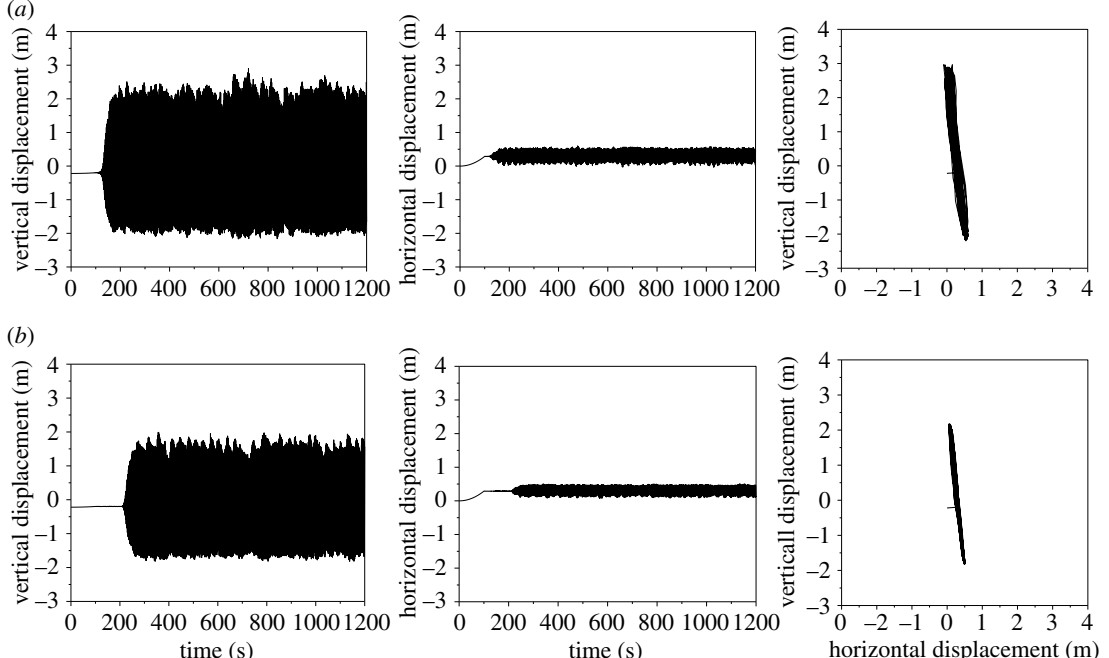

**Figure 9.** Displacement time histories and motion traces at the mid-span of sub-conductor 1 in the iced quad bundle conductor line with different aerodynamic coefficients (ice thickness = 12 mm, initial wind attack angle = 175°, wind velocity = 12 m s$^{-1}$). (*a*) Aerodynamic coefficients by wind tunnel test and (*b*) aerodynamic coefficients by numerical simulation.

galloping takes place when the initial wind attack angle locates in the ranges of 265–280° and 310–350°. It is seen from figure 7 that the Den Hartog coefficients of the four sub-conductors in the range of 170–185° are negative, but those of sub-conductors 3 and 4 in the range of 265–350° are positive. This may be the reason why the galloping takes place when the initial wind attack angle is in the range of 170–185° and no galloping takes place when the initial wind attack angle is in the ranges of 265–280° and 310–350°.

The dynamic responses of the line with an initial wind attack angle of 175° based on the aerodynamic coefficients by numerical simulation and test are shown in figure 9 and the power spectra of the displacements of sub-conductor 1 are shown in figure 10. Although there are some deviations of the vibration amplitudes and motion traces based on the aerodynamic coefficients by the two methods, the galloping modes are both vertical galloping and the galloping frequencies are nearly the same. It is demonstrated that the galloping features of iced lines using the tested and simulated coefficients are similar, especially the galloping mode and frequency, which can be used to guide the design of anti-galloping technology and devices. Therefore, the deviation is acceptable and the simulated coefficients can be used to investigate the galloping.

## 2.4. Numerical simulation of aerodynamic coefficients

The aerodynamic coefficients of each sub-conductor of the quad bundle conductor with 12 mm ice discussed in §2.1 varying with wind attack angle under wind velocity of 12 m s$^{-1}$ are shown in figure 11. It can be seen that the aerodynamic coefficients of different sub-conductors are obviously

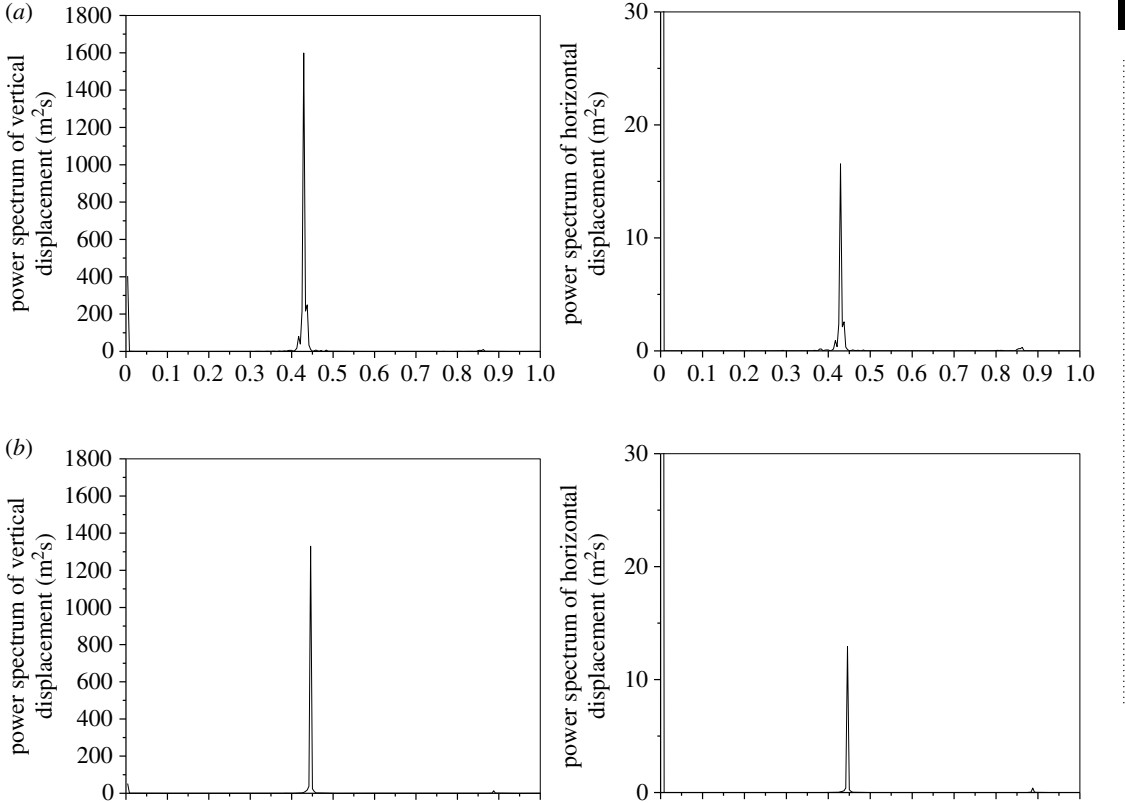

**Figure 10.** Power spectra of displacements at the mid-span of sub-conductor 1 in the iced quad bundle conductor line with different aerodynamic coefficients (ice thickness = 12 mm, initial wind attack angle = 175°, wind velocity = 12 m s$^{-1}$). (*a*) Aerodynamic coefficients by wind tunnel test and (*b*) aerodynamic coefficients by numerical simulation.

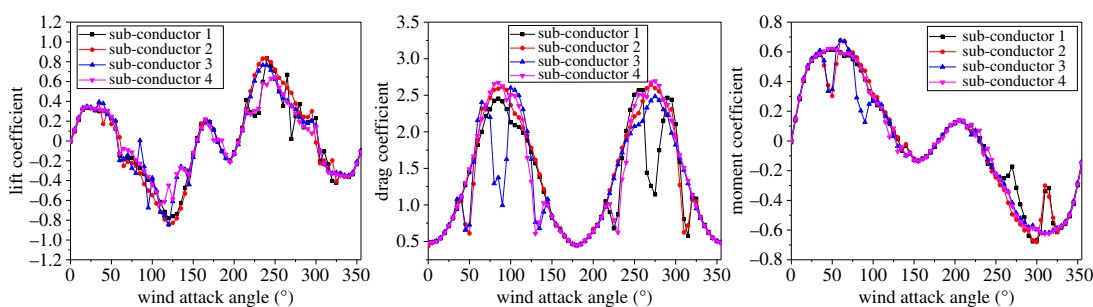

**Figure 11.** Aerodynamic coefficients of different sub-conductors varying with wind attack angle in the iced quad bundle conductor. (Ice thickness: 12 mm; wind velocity: 12 m s$^{-1}$).

different at some wind attack angles due to the wake influence of the windward sub-conductors on the leeward sub-conductors.

Furthermore, the ice thickness, ice accretion angle, wind velocity, conductor type (diameter) and spacing between adjacent sub-conductors may affect the aerodynamic characteristics of the iced quad bundle conductors. The curves of the lift, the drag and the torsional moment coefficients of the iced quad bundle conductors versus wind attack angle under different parameters are shown in figure 12. It is seen from figure 12*a* that all the values of the lift, drag and moment coefficients increase with ice thickness, but the changing law of each coefficient is similar in the cases of different ice thickness. The wake influence zones of the windward sub-conductors on the leeward sub-conductors are different in the cases of different ice accretion angles, as shown in figure 12*b*. For example, the aerodynamic coefficients of sub-conductor 1 are influenced by the wake interference around the wind attack angles of 195°, 240° and 285° for the ice accretion angle of 15° while around the wind attack angles of 255°,

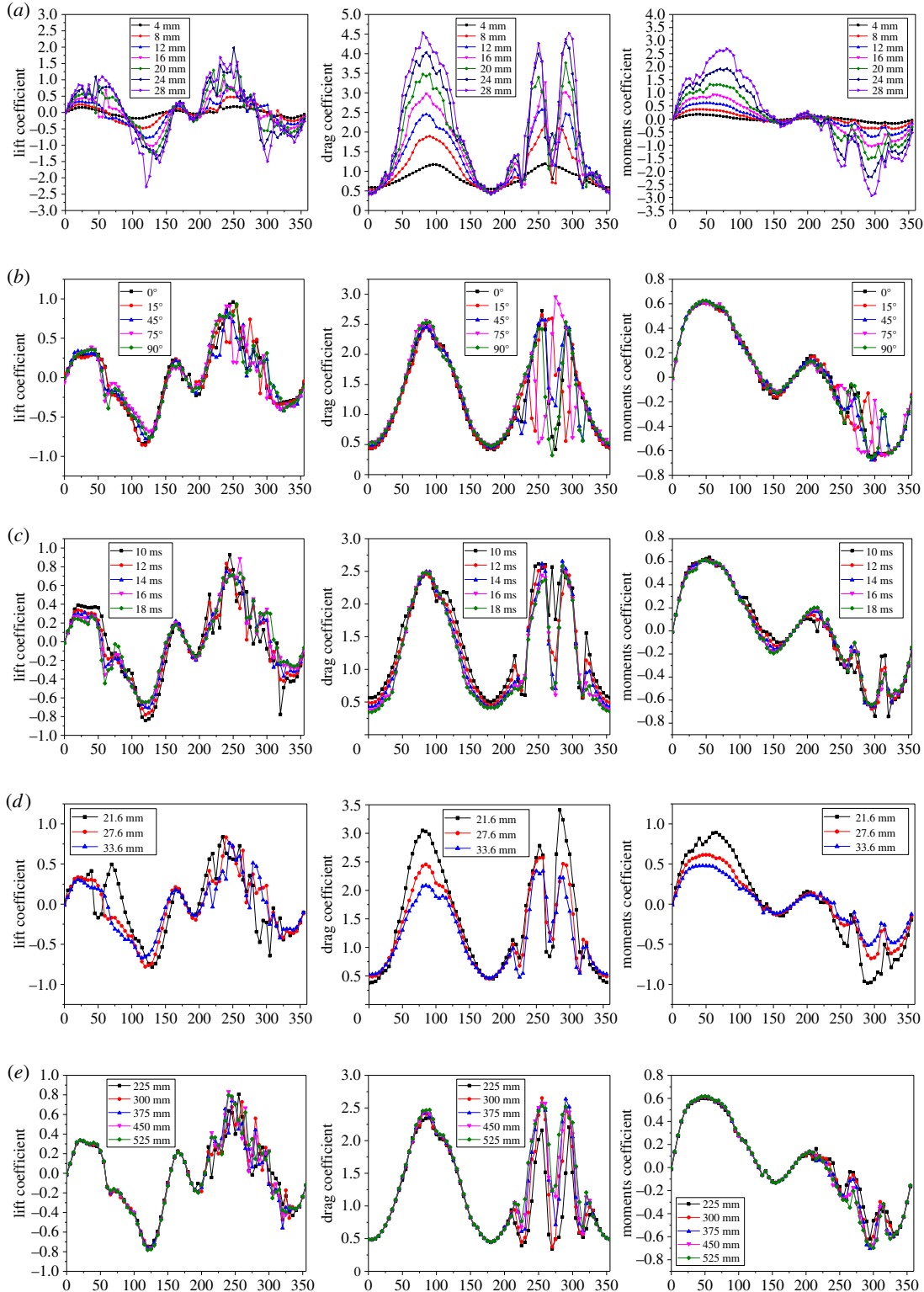

**Figure 12.** Aerodynamic coefficients of sub-conductor 1 of the iced quad bundle conductors varying with wind attack angle in the cases of (*a*) different ice thickness; (*b*) different ice accretion angle; (*c*) different wind velocity; (*d*) different conductor type and (*e*) different bundle spacing.

300° and 345° for the ice accretion angle of 75°. It is seen from figure 12*c* that wind velocity has little influence on the aerodynamic coefficients of the iced conductor. In addition, the conductors LGJ-240/30, LGJ-400/50 and LGJ 630/45 with diameter of 21.6, 27.6 and 33.6 mm, respectively, are chosen to investigate the effect of the conductor type. As shown in figure 12*d*, the absolute values of the drag

and moment coefficients increase obviously with the diameter of the sub-conductor while the absolute values of lift coefficients increase slightly. It can be seen from figure 12*e* that the wake interference becomes more apparent with the decrease of bundle spacing and the differences of lift, drag and moment coefficients are more obvious around the wind attack angles of 225°, 270° and 315°.

# 3. Prediction model for aerodynamic coefficients

To develop a prediction model for aerodynamic coefficients of the iced quad bundle conductors, two datasets of the aerodynamic coefficients of the iced quad bundle conductors with different bundle spacing, ice accretion angle, ice thickness, wind velocity and conductor type, must be set up. The datasets consist of the aerodynamic data obtained by the numerical simulation at the designed sampling points using the LHS method. Based on the constructed datasets and machine learning method, the prediction model is then created, trained and tested.

## 3.1. Design of data sampling

### 3.1.1. Latin hypercube sampling

The selection of sampling points, namely the design of experiment (DOE), has significant influence on the performance of the prediction model constructed by machine learning method. The purpose of the DOE is to reflect the information of design space with limited sampling points and reduce the experimental cost. The LHS which adopts space-filling design and avoids the aggregation of points is a popular DOE method. Assuming that there are $n_v$ design variables and $n_s$ sampling points, the range of each variable is divided into $n_s$ parts and then $n_v^{ns}$ zones can be acquired in the whole design space. Subsequently, the $n_s$ sample points are spread randomly in the $n_v^{ns}$ zones with the satisfaction of two conditions: (i) each sampling point is placed inside the zone randomly; (ii) each part has just one sampling point when projected on the axis of variable. Assuming the range of each variable to be [0, 1], the sampling points can be generated by the following formulation:

$$x_j^{(i)} = \frac{\pi_j^{(i)} + U_j^{(i)}}{n_s}, \quad 1 \le j \le n_v, 1 \le i \le n_s, \tag{3.1}$$

where *i* represents the *i*th sampling point, *j* represents the *j*th design variable, $\pi_j$ is a random permutation of {0, 1, ..., $n_s - 1$} and is a random number in [0, 1].

The combination of sampling points generated randomly by the LHS usually may not be the best choice for the design space, and thus the optimal LHS methods are developed. Morris & Mitchell [21] proposed an optimal LHS method, which maximizes the minimum distance among points and sets the points at centres of the sampling intervals, and this method is employed in this paper.

### 3.1.2. Sampling points for the training prediction model

Based on the analysis of §2.4, the lift, drag and moment coefficients of each sub-conductor of an iced quad bundle conductor with crescent-shaped ice depend on five factors, including bundle spacing, ice accretion angle, ice thickness, wind velocity and conductor type. These parameters are chosen as the design variables for the sampling, and the ranges of these variables are shown in table 1. The bundle spacing is set from 250 to 550 mm and the diameter of the conductor characterizing the conductor type is set from 21.6 to 33.6 mm. The accreted ice is the most important factor influencing the aerodynamic loads of the iced conductors and galloping of the transmission lines. Only crescent-shaped ice is taken into account in this paper and it is characterized by the thickness and ice accretion angle. The range of ice thickness is set to be 4–28 mm and the ice accretion angle is in the range of −90° to 90°. The wind velocity of inducing the galloping of iced quad bundle conductor lines is over 10 m s$^{-1}$ usually [6], and thus the wind velocities ranging from 10 to 18 m s$^{-1}$ are considered.

According to the ranges of design variables, 18 sampling points for training the prediction model are determined by means of the optimal LHS method. The aerodynamic coefficients of the iced quad bundle conductors at the sampling points as shown in table 2 are calculated by the developed software discussed in §2.1. There are 72 samples for each iced sub-conductor, which depict the aerodynamic characteristics of the iced sub-conductor under the wind attack angles of 0–360° with an increment of 5°. Totally there are

**Table 1.** Ranges of the design variables.

| design variables | symbol | unit | ranges |
|---|---|---|---|
| bundle spacing | BS | mm | 250–550 |
| ice accretion angle | IA | ° | −90–90 |
| wind velocity | WV | m s$^{-1}$ | 10–18 |
| ice thickness | IT | mm | 4–28 |
| diameter of conductor | DC | mm | 21.6–33.6 |

**Table 2.** Sampling points for training of prediction model.

| no. | bundle spacing (mm) | diameter of conductor (mm) | wind velocity (m s$^{-1}$) | ice thickness (mm) | ice accretion angle (°) |
|---|---|---|---|---|---|
| 1 | 467 | 31.3 | 12.4 | 22.0 | 15 |
| 2 | 450 | 28.6 | 17.3 | 10.0 | −15 |
| 3 | 500 | 23.3 | 16.0 | 23.3 | 65 |
| 4 | 250 | 32.6 | 12.9 | 16.7 | −85 |
| 5 | 333 | 31.9 | 10.7 | 7.3 | −35 |
| 6 | 283 | 27.9 | 13.8 | 20.7 | 25 |
| 7 | 517 | 24.6 | 11.6 | 27.3 | −5 |
| 8 | 267 | 27.6 | 15.1 | 14.0 | −75 |
| 9 | 317 | 25.3 | 12.0 | 12.7 | 55 |
| 10 | 233 | 33.3 | 11.1 | 26.0 | −65 |
| 11 | 400 | 25.9 | 14.2 | 6.0 | 5 |
| 12 | 383 | 21.9 | 17.8 | 11.3 | 85 |
| 13 | 417 | 23.9 | 10.2 | 15.3 | 35 |
| 14 | 367 | 29.3 | 16.4 | 18.0 | 45 |
| 15 | 300 | 22.6 | 13.3 | 8.7 | −25 |
| 16 | 350 | 26.6 | 16.9 | 4.7 | −55 |
| 17 | 433 | 29.9 | 15.6 | 19.3 | −45 |
| 18 | 483 | 30.6 | 14.7 | 24.7 | 75 |

288 samples for one quad bundle conductor consisting of four sub-conductors and 5184 samples for the 18 sampling points, which are aggregated as the training dataset for the prediction model.

### 3.1.3. Sampling points for the testing prediction model

In order to evaluate the generalization ability and prediction accuracy of the prediction model for aerodynamic coefficients of iced quad bundle conductors, the other six sampling points as shown in table 3 for testing the model are generated using the optimal LHS method. Similarly, there are 288 samples at each testing sampling point to characterize the aerodynamic characteristics of the four sub-conductors and totally 1728 samples for the six sampling points in the testing dataset.

## 3.2. Creation and training of prediction model

### 3.2.1. Extra-trees algorithm

As a new developed ensemble algorithm in machine learning, the extra-trees algorithm proposed by Geurts *et al*. [22] has attracted the attention of many researchers in recent years. The principle of the

**Table 3.** Sampling points for testing of prediction model.

| no. | bundle spacing (mm) | diameter of conductor (mm) | wind velocity (m s$^{-1}$) | ice thickness (mm) | ice accretion angle (°) |
|---|---|---|---|---|---|
| 1 | 500 | 24.6 | 10.7 | 22.0 | 15 |
| 2 | 250 | 28.6 | 17.3 | 18.0 | 75 |
| 3 | 300 | 32.6 | 14.7 | 26.0 | −15 |
| 4 | 400 | 30.6 | 12.0 | 14.0 | −45 |
| 5 | 450 | 22.6 | 16.0 | 10.0 | −75 |
| 6 | 350 | 26.6 | 13.3 | 6.0 | 45 |

algorithm, which is called bagging, is to integrate many weak classification and regression decision trees [23] to generate a strong model. For the splitting node $m$ of a decision tree, the samples are partitioned into subsets $Q_m^l$ and $Q_m^r$ based on the splitting rule $\theta = (j, t_m)$ consisting of a feature $j$ and its splitting threshold, which can be expressed as

$$Q_m^l(\theta) = \{(x,y)|x_j <= t_m\} \tag{3.2}$$

and

$$Q_m^r(\theta) = Q_m \backslash Q_m^l(\theta), \tag{3.3}$$

where $x$ and $y$ are the input and output data of the samples, respectively.

In the decision-tree algorithm containing single tree, the splitting rule is determined by minimizing the impurity, which is the function of $Q_m^l$ and $Q_m^r$. However, the splitting feature and corresponding threshold in the splitting rule are both selected at random fully in the decision trees of the extra-trees algorithm. The final predicted results are given by the group decision, i.e. the majority vote in the classification problems and arithmetic average in the regression problems, which also reduce the high variance, improve the accuracy and prevent the overfitting simultaneously. Therefore, the advantages of the extra-trees algorithm, including high accuracy, resistance against overfitting, swarm intelligence and convenience to realize, make it one of the most popular machine learning methods.

## 3.2.2. Model creation and training

By means of the extra-trees algorithm, the prediction model for aerodynamic coefficients of iced quad bundle conductors is created. The five design variables, i.e. the bundle spacing (BS), ice accretion angle (IA), wind velocity (WV), ice thickness (IT) and diameter of conductor (DC), are set as the input of the model. On the other hand, the variation of the lift, drag and moment coefficients of each sub-conductor of an iced quad bundle conductor versus wind attack angle is set as the output. It is noted that during the creation and training of the prediction model whose structure is as shown in figure 13, the number of each sub-conductor (NC) ranging from 1 to 4 and wind attack angle (WA) ranging from 0° to 360° with an interval of 5° are also set as input variables and the corresponding aerodynamic coefficients of each sub-conductor at the wind attack angle are set as output variables. In the final integrated prediction model, the aerodynamic coefficients of each sub-conductor versus the discrete wind attack angle incrementally changing in the range of 0–360° are output.

The training dataset with 5184 samples set up in §3.1.2 is adopted to train the prediction model. Three parameters in the extra-trees algorithm, i.e. the number of parallel decision trees (NT), the maximum number of splitting features (MF) and the minimum number of samples required to split a node (MS), need to be determined. The results of trial and error indicate that the predicted accuracy increases with the number of decision trees and the optimal NT is set to be 100 finally. The parameter MF is set to be the dimension of the input variables usually for regression problems according to the suggestion [22]. Moreover, it is found that the model performance does not have obvious enhancement with the increase of parameter MS and thus a default of 2 is set for the parameter. Due to the parallelization capability of the algorithm, the decision trees grow simultaneously and the created prediction model is trained quickly, and this process takes only a few minutes. The comparison between the predicted aerodynamic coefficients by the prediction model and the true values by the numerical simulation is

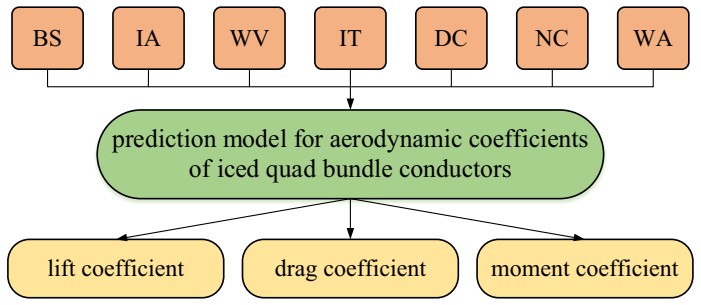

**Figure 13.** Structure of prediction model for aerodynamic coefficients of iced quad bundle conductors.

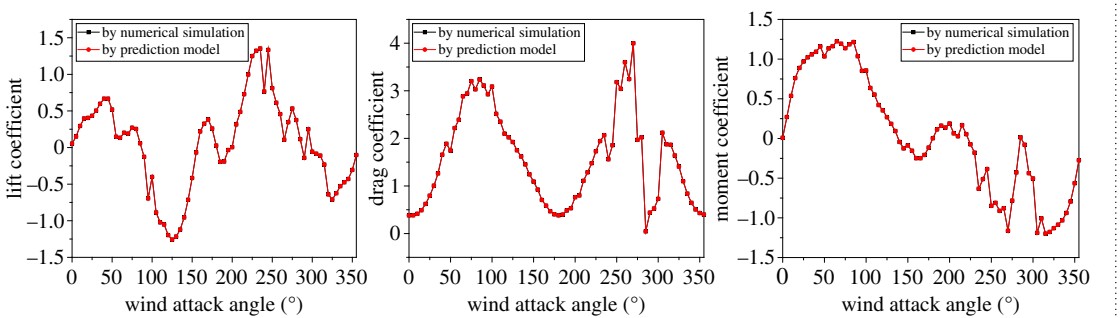

**Figure 14.** Comparison of aerodynamic coefficients of sub-conductor 1 varying with wind attack angle by numerical simulation and prediction model at training sampling point no. 1.

shown typically in figure 14. It can be seen that the predicted coefficients are consistent with the true values and the prediction model has studied fully from the training dataset.

## 3.3. Testing of the prediction model

The generalization ability of the trained prediction model is tested by employing the testing dataset containing six sampling points. The comparison of the aerodynamic coefficients of the four sub-conductors varying with wind attack angle by numerical simulation and prediction model at testing sampling point no. 1 is shown in figure 15. It can be observed that the predicted aerodynamic coefficients are consistent with those determined by the numerical simulation. Although there are some deviations in some zones with sharp change, the changing laws of the predicted lift, drag and moment coefficients are very similar with those by the numerical simulation.

The mean absolute error ($MAE$), mean squared error ($MSE$) and coefficient of determination ($R^2$) are adopted to evaluate the generalization ability of the trained prediction model on the testing dataset. These evaluation indices are defined, respectively, as

$$MAE = \frac{1}{N}\sum_{i=1}^{N}|y_i - y_i'|, \tag{3.4}$$

$$MSE = \frac{1}{N}\sum_{i=1}^{N}(y_i - y_i')^2 \tag{3.5}$$

and

$$R^2 = 1 - \sum_{i=1}^{N}(y_i - y_i')^2 / \sum_{i=1}^{N}(y_i - \bar{y})^2, \tag{3.6}$$

where $y_i$ denotes the true value, $y_i'$ denotes the predicted value, $\bar{y}$ denotes the average of true values and $N$ is the number of testing samples. If the predicted results are excellent, the $MAE$ and $MSE$ are close to 0 while the $R^2$ is close to 1.

The evaluation indices of the prediction model by the testing dataset are calculated and listed in table 4. As shown in the table, the average $MAE$ values of the lift, drag and moment coefficients are 0.12, 0.20 and 0.11, respectively, while the average $MSE$ values are 0.03, 0.11 and 0.03. In addition, all of the average $R^2$ values of the three aerodynamic coefficients are greater than 0.87, which reveals that

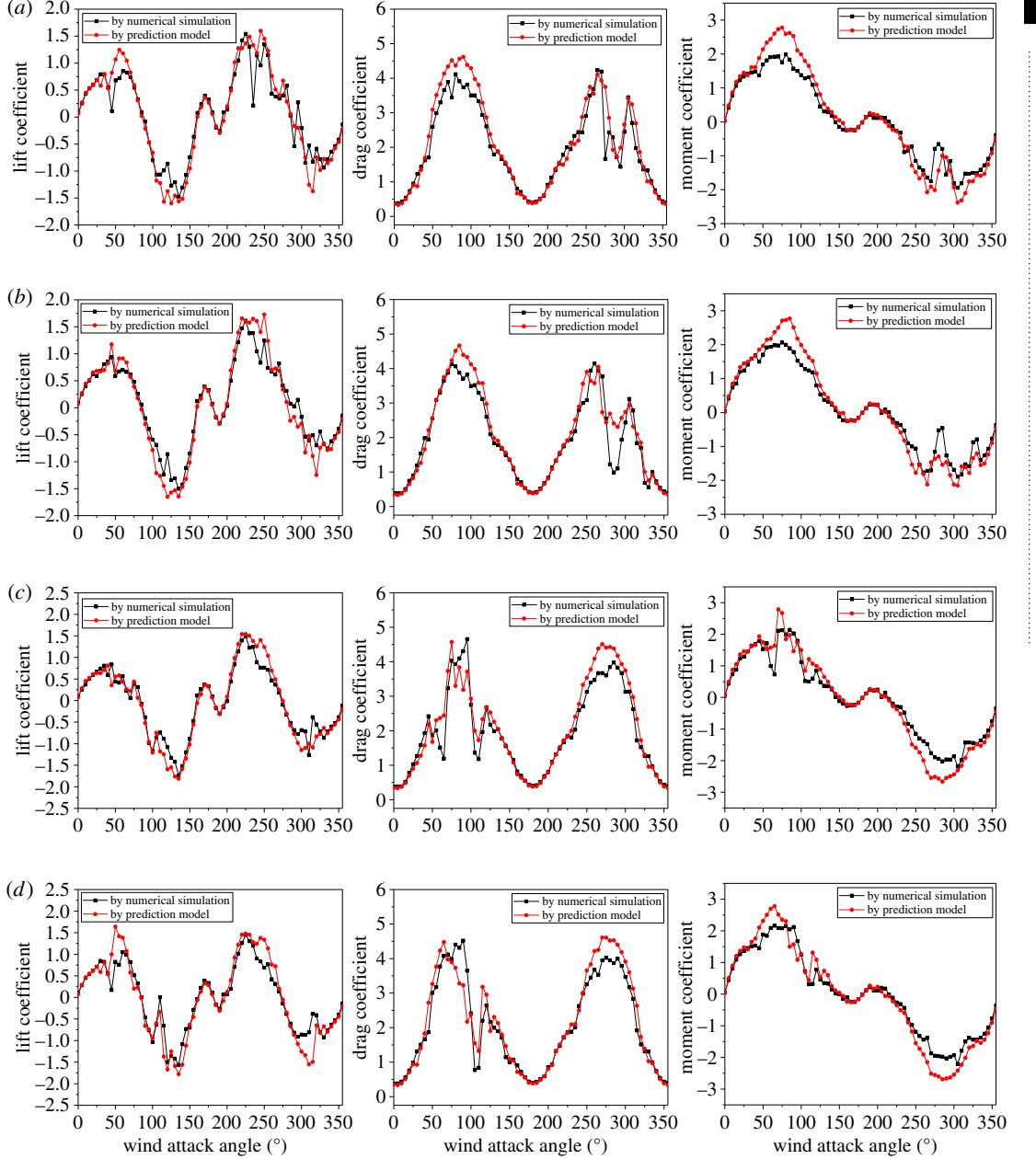

**Figure 15.** Comparison of aerodynamic coefficients of sub-conductors varying with wind attack angle by numerical simulation and prediction model at testing sampling point no. 1: (*a*) sub-conductor 1; (*b*) sub-conductor 2; (*c*) sub-conductor 3 and (*d*) sub-conductor 4.

**Table 4.** Evaluation indices of prediction model by testing dataset.

| no. | lift coefficient | | | drag coefficient | | | moment coefficient | | |
|---|---|---|---|---|---|---|---|---|---|
| | *MAE* | *MSE* | $R^2$ | *MAE* | *MSE* | $R^2$ | *MAE* | *MSE* | $R^2$ |
| 1 | 0.18 | 0.072 | 0.87 | 0.28 | 0.17 | 0.88 | 0.27 | 0.13 | 0.91 |
| 2 | 0.16 | 0.046 | 0.79 | 0.21 | 0.12 | 0.86 | 0.08 | 0.013 | 0.96 |
| 3 | 0.17 | 0.055 | 0.87 | 0.33 | 0.24 | 0.78 | 0.16 | 0.05 | 0.93 |
| 4 | 0.092 | 0.019 | 0.88 | 0.16 | 0.063 | 0.89 | 0.067 | 0.0095 | 0.94 |
| 5 | 0.079 | 0.014 | 0.90 | 0.15 | 0.062 | 0.89 | 0.049 | 0.0042 | 0.97 |
| 6 | 0.033 | 0.0017 | 0.95 | 0.062 | 0.011 | 0.94 | 0.013 | 0.00027 | 0.99 |
| average | 0.12 | 0.03 | 0.88 | 0.20 | 0.11 | 0.87 | 0.11 | 0.03 | 0.95 |

(a)

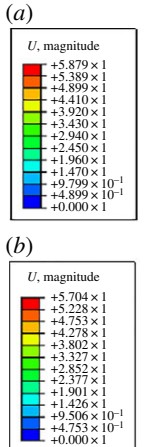

(b)

**Figure 16.** Galloping shapes at a moment of iced quad bundle conductor line with different aerodynamic coefficients. (a) Aerodynamic coefficients by numerical simulation and (b) aerodynamic coefficients by prediction model.

over 87% of the aerodynamic coefficients in the testing dataset are correctly predicted. Moreover, the average time for predicting the curves of the aerodynamic coefficients of an iced quad bundle conductor varying with wind attack angle by means of the trained model is only about 10 s, which is very quick compared with the average 30 h by numerical simulation. In summary, the prediction model can be used to predict the aerodynamic coefficients of iced quad bundle conductors quickly and effectively.

# 4. Validation and application of prediction model

To validate the prediction model further, the aerodynamic coefficients predicted by the model and those by the numerical simulation are both applied to simulate the galloping of a typical iced transmission line and the extracted galloping features depending on the aerodynamic coefficients by the two methods are compared. Moreover, the influences of the bundle spacing and conductor type on the galloping features, which are rarely considered in the previous work, are also investigated by applying the prediction model.

## 4.1. Galloping features based on aerodynamic coefficients by the two methods

The research group of the authors proposed a numerical simulation method based on the ABAQUS software for the galloping of iced transmission lines. The sub-conductors are discretized by cable element with torsional degree-of-freedom, and the aerodynamic forces on each element depend on the wind attack angle which is related to the motion state of the element at every moment during galloping. Therefore, an element which shares the same nodes with each parallel cable element is added to acquire the real-time motion state. Based on the aerodynamic coefficients, the aerodynamic forces on each cable element can be determined and applied using the user-defined subroutine UEL. More details of the method can be found in the work of Hu *et al.* [6].

The finite-element model of a typical quad bundle conductor line with span length of 200 m is shown in figure 8. The longitudinal direction of the line is along the $x$-axis, the horizontal direction along the $y$-axis and the vertical along the $z$-axis. The conductor type is LGJ-400/50, the initial conductor tension is 27.09 kN and the bundle spacing is 450 mm. The spacers are simplified as square frames with mass of 7.9 kg and the distance between each other is 50 m. The thickness of the crescent-shaped ice accreted on the sub-conductors is 12 mm and the ice accretion angle is 45°. The wind velocity is 14 m s$^{-1}$ and the initial wind attack angle is 45°.

The aerodynamic coefficients obtained by the prediction model and numerical simulation are, respectively, used to simulate the galloping of the quad bundle conductor line. The galloping shapes of the line at a typical moment are shown in figure 16, from which it is seen that the vertical vibration modes (VM) of the line with different aerodynamic coefficients are both one-loop and the maximum vibration amplitudes occur at the mid-span of the line. Thus, the displacement time histories and motion traces at the mid-span of sub-conductor 1 of the iced quad bundle conductor line are extracted and shown in figure 17. Although the inducing processes of the two galloping phenomena are

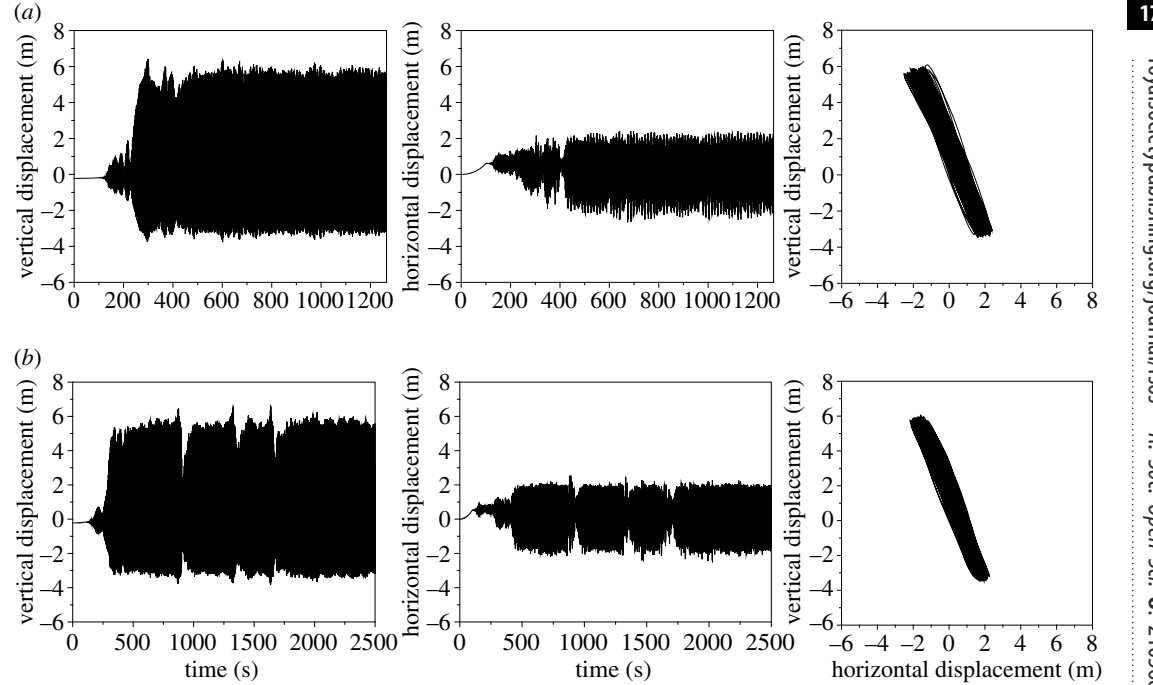

**Figure 17.** Displacement time histories and motion traces at the mid-span of sub-conductor 1 in the iced quad bundle conductor line with different aerodynamic coefficients. (*a*) Aerodynamic coefficients by numerical simulation and (*b*) aerodynamic coefficients by prediction model.

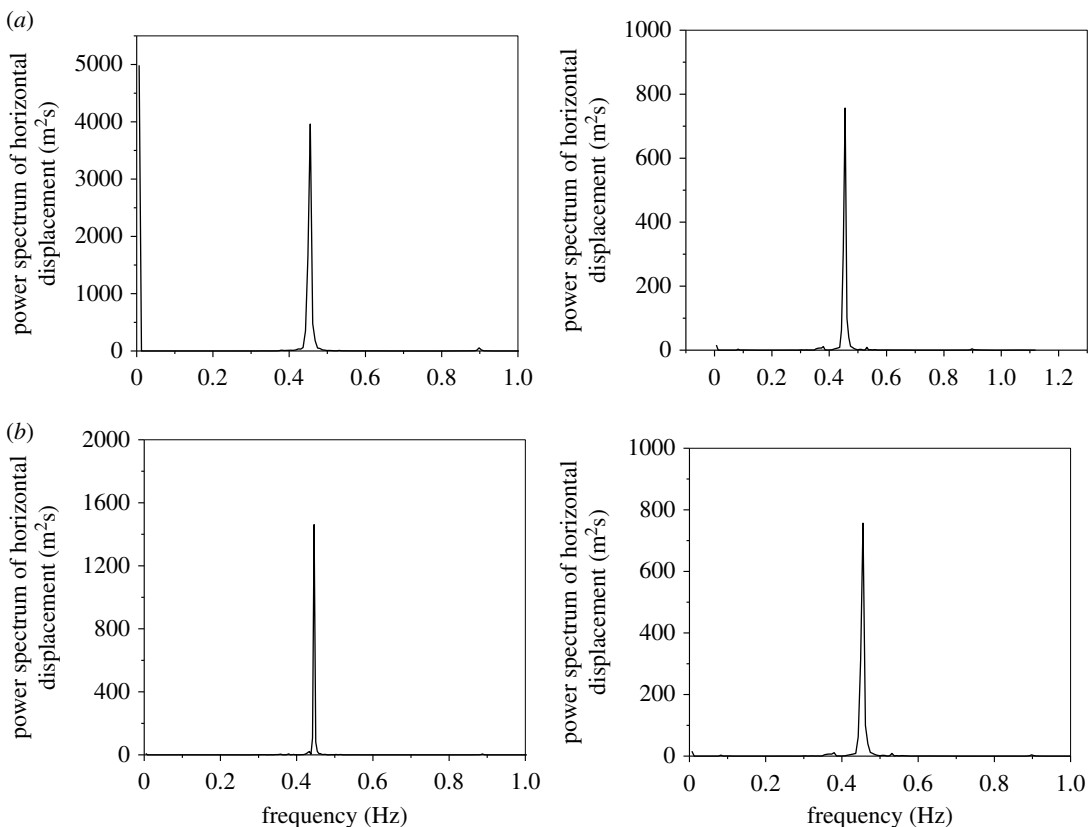

**Figure 18.** Power spectra of displacements at the mid-span of sub-conductor 1 in the iced quad bundle conductor line with different aerodynamic coefficients. (*a*) Aerodynamic coefficients by numerical simulation and (*b*) aerodynamic coefficients by prediction model.

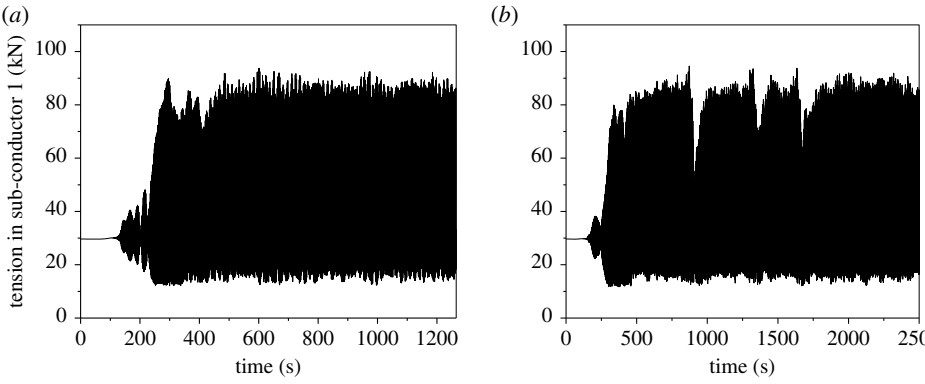

**Figure 19.** Tension time histories of sub-conductor 1 in the iced quad bundle conductor line with different aerodynamic coefficients. (*a*) Aerodynamic coefficients by numerical simulation and (*b*) aerodynamic coefficients by prediction model.

**Table 5.** Galloping features of iced quad bundle conductor lines with different bundle spacing.

| aerodynamic coefficients | bundle spacing (mm) | VM | VA (m) | HA (m) | VF (Hz) | HF (Hz) | MT (kN) |
|---|---|---|---|---|---|---|---|
| by numerical simulation | 375 | one-loop | 7.78 | 3.20 | 0.44 | 0.44 | 91.9 |
| | 450 | one-loop | 8.72 | 3.83 | 0.46 | 0.46 | 93.9 |
| | 525 | one-loop | 8.57 | 3.36 | 0.43 | 0.45 | 103.5 |
| by prediction model | 375 | one-loop | 7.83 | 2.51 | 0.44 | 0.43 | 89.6 |
| | 450 | one-loop | 8.52 | 3.51 | 0.45 | 0.45 | 94.5 |
| | 525 | one-loop | 8.67 | 3.29 | 0.43 | 0.43 | 113.3 |

slightly different, the vertical vibration amplitude (VA), horizontal vibration amplitude (HA) and the motion trace are almost the same when the galloping achieves steady state. Furthermore, the power spectra of the displacements are analysed to investigate the galloping frequency, as shown in figure 18. Both the vertical galloping frequency (VF) and horizontal galloping frequency (HF) of the line are the same, and they are, respectively, 0.46 and 0.45 Hz based on the aerodynamic coefficients by the numerical simulation and the prediction model. In addition, the time histories of conductor tension of these two cases are shown in figure 19. The maximum conductor tensions (MT) of sub-conductor 1 of the line based on the aerodynamic coefficients by the numerical simulation and the prediction model are 93.94 and 94.50 kN, respectively. It is seen that the galloping of the iced quad bundle conductor transmission line based on the aerodynamic coefficients predicted by the developed model are very similar to those on the aerodynamic coefficients obtained by the numerical simulation method. Therefore, the prediction model can be used for the investigation of galloping of iced quad bundle conductor transmission lines.

## 4.2. Galloping features of quad bundle conductors with different bundle spacing and conductors

It is implied in §2.4 that the bundle spacing influences the aerodynamic coefficients of iced sub-conductors in the zones of wake interference, and it may, in turn, affect the galloping behaviour, which is further studied in this section.

The galloping of iced quad bundle conductor lines with bundle spacing of 375, 450 and 525 mm, respectively, is numerically investigated, and other parameters are set to be the same as those in §4.1. Comparison of the extracted six galloping features of the iced quad bundle conductor lines with different bundle spacing based on the aerodynamic coefficients by the numerical simulation and the prediction model are shown in table 5. It is seen that all the galloping features based on the aerodynamic coefficients by the numerical simulation and the prediction model, respectively, are very

**Table 6.** Galloping features of iced quad bundle conductor lines with different conductors.

| aerodynamic coefficients | conductor type | VM | VA (m) | HA (m) | VF (Hz) | HF (Hz) | MT (kN) |
|---|---|---|---|---|---|---|---|
| by numerical simulation | LGJ-240/30 | one-loop | 8.46 | 5.10 | 0.53 | 0.53 | 81.1 |
| | LGJ-400/50 | one-loop | 8.72 | 3.83 | 0.46 | 0.46 | 93.9 |
| | LGJ-630/45 | one-loop | 4.75 | 1.31 | 0.45 | 0.45 | 122.7 |
| by prediction model | LGJ-240/30 | one-loop | 9.02 | 5.45 | 0.55 | 0.55 | 79.5 |
| | LGJ-400/50 | one-loop | 8.52 | 3.51 | 0.45 | 0.45 | 94.5 |
| | LGJ-630/45 | one-loop | 5.12 | 1.66 | 0.46 | 0.46 | 126.0 |

closed. Moreover, one-loop galloping is induced in all the cases with different bundle spacing and its effect on galloping frequencies are very small. In addition, the vibration amplitudes in vertical and horizontal directions and the maximum conductor tension increase slightly with the bundle spacing. It is concluded that in the discussed range, the effect of bundle spacing on the galloping features is limited.

The galloping of iced quad bundle conductor lines with conductor types of LGJ-240/30, LGJ-400/50 and LGJ-630/45, respectively, is simulated. The initial stress of the lines with the three types of conductors is set to be 60 MPa, i.e. the initial conductor tensions are 21.99, 27.93 and 53.68 kN, respectively. The ice and wind parameters are kept the same as those discussed in §4.1. Comparison of the extracted six galloping features of the iced quad bundle conductor lines with different conductors based on the aerodynamic coefficients by the numerical simulation and the prediction model are shown in table 6. It is seen that all the galloping features based on the aerodynamic coefficients by the numerical simulation and the prediction model, respectively, are very close. The vertical galloping modes of the lines with three types of conductors are all one-loop and the galloping frequencies in vertical and horizontal directions are all close to the one-order natural vertical frequencies of the lines. The vibration amplitudes in both directions decrease with the diameter of the conductors. Moreover, the maximum conductor tensions of the conductors LGJ-240/30, LGJ-400/50 and LGJ-630/45 during galloping are, respectively, 3.69, 3.36 and 2.29 times of the initial tensions. Especially, the maximum tension of conductor LGJ-240/30, i.e. 81.1 or 79.5 kN, is greater than its strength of 75.62 kN.

## 5. Conclusion

The aerodynamic characteristics of iced quad bundle conductors are simulated by numerical method, and parameter study is carried out to identify the effect of various factors on the aerodynamic coefficients. By means of the machine learning method, a prediction model for aerodynamic coefficients of iced quad bundle conductors is created, trained and tested. Furthermore, the developed model is validated and applied to predict the coefficients for simulating the galloping. It is concluded that:

(i) The changing laws of the lift, drag and moment coefficients of iced quad bundle conductors obtained by the numerical simulation are consistent with those measured by the wind tunnel tests.

(ii) Parameter study shows that the bundle spacing and ice accretion angle only affect the aerodynamic coefficients of the iced sub-conductor in the zones of wake interference while the wind velocity, ice thickness and conductor type affect the aerodynamic coefficients in the whole range of wind attack angles.

(iii) The developed prediction model can determine the aerodynamic coefficients of iced sub-conductors quickly and effectively, which is demonstrated by the $R^2$ value of each coefficient being greater than 0.87 and small values of $MAE$ and $MSE$ in the testing.

(iv) Analysis on the galloping features of iced quad bundle conductor transmission lines indicates that the aerodynamic coefficients determined by the prediction model can be used to investigate the galloping of iced transmission lines. The bundle spacing has slight influence on the galloping and the influence of the conductor type is obvious.

(v) The application range of the prediction model created in this paper is limited. However, it provides a possible way to develop a prediction model for the aerodynamic coefficients of iced transmission lines with a wide range of parameters in the future.

Data accessibility. Our data are deposited at Dryad Digital Repository: https://doi.org/10.5061/dryad.dv41ns1xt [24].

Authors' contributions. Z.M., B.Y., H.Y. and D.C. carried out the numerical simulation and wind tunnel tests, developed the prediction model and simulated the galloping; G.H. developed the simulation software for aerodynamic characteristics of iced conductors based on the computational fluid dynamics.

Competing interests. We declare we have no competing interests.

Funding. This work is sponsored by the National Natural Science Foundation of China (grant no. 11572060).

Acknowledgements. We thank Yongqiang Zhu and Yingbo Gao for their help in numerical simulation and Jiaqiong Liu for his advice in the details of this paper. Besides, we also thank editors and anonymous reviewers for their helpful suggestions on early versions of this manuscript.

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
