## [Peer Review File · Royal Society Open Science]

Review History

RSOS-210568.R0 (Original submission)

Review form: Reviewer 1

Is the manuscript scientifically sound in its present form?

Yes

Are the interpretations and conclusions justified by the results?

Yes

Is the language acceptable?

Yes

Do you have any ethical concerns with this paper?

No

Have you any concerns about statistical analyses in this paper?

No

Recommendation?

Major revision is needed (please make suggestions in comments)

Comments to the Author(s)

It is interesting to establish a prediction model for aerodynamic coefficients of iced quad bundle conductors. The authors created datasets by using the Latin hypercube sampling (LHS) and numerical simulation. The extra-trees algorithm was also adopted to realize the mapping from the number of sub-conductors, wind attack angle, bundle spacing, ice accretion angle, ice thickness, wind velocity, and diameter of conductor to the aerodynamic coefficients of iced quad bundle conductors. The overall idea is good, but there are still some issues that need attention, as listed below:

1. In figure 6, it seems that the results of numerical simulation are not in good agreement with the wind tunnel test results. The lift coefficient curves of the two ways do not overlap when the wind attack angle is less than 50 degrees greater than 310 degrees. Previous studies have shown that the lift coefficient of the crescent-shaped iced conductors changes dramatically in the above-mentioned range of wind attack angle, so it needs to be treated with extreme caution. Such simulation results are used as training data, making the accuracy of the prediction results questionable. It is suggested to validate the in-house simulation software against the commercial one, i.e., Fluent, before using the simulation results to train a machine learning model.
2. Please compare the computational cost of the numerical simulation and the prediction model. Since the aerodynamic coefficients of iced conductors could be well simulated by the in-house CFD software, it reduces the importance of developing the machine learning based prediction model. Please discuss and clarify this issue.

Review form: Reviewer 2

Is the manuscript scientifically sound in its present form?

No

Are the interpretations and conclusions justified by the results?

No

Is the language acceptable?

Yes

Do you have any ethical concerns with this paper?

No

Have you any concerns about statistical analyses in this paper?

No

Recommendation?

Major revision is needed (please make suggestions in comments)

Comments to the Author(s)

Review of the manuscript RSOS-210568, "Prediction model for aerodynamic coefficients of iced quad bundle conductors based on machine learning method" by Zheyue Mou, Bo Yan, Hanxu Yang, Daoda Cai, Guizao Huang, submitted for publication in Royal Society Open Science.

The paper proposes a prediction model based on machine learning to evaluate the aerodynamic coefficients of a quad bundle of iced conductors. Description of the numerical model which is used to build the dataset is first given. Then the prediction model, based on the work proposed by Geurts et al. [22], is realized and tested. Comparison of the galloping features of conductors, as evaluated using coefficients which are outcomes of both numerical and prediction procedures, is done.

The paper shows two distinct souls: on the one hand the first piece, which belongs to the field of mechanics, is interesting but it is a summary of other jobs already published by some of the Authors [12]; on the other hand, the second piece (Section 3), is the actual core of the paper and describes the machine learning algorithm. It belongs to the information technology field and, in the context of the problem, appears as pointless. In particular, it is not clear (or not clearly described) the necessity of creating such a machine learning structure to (not exactly) reproduce outcomes which are available by a mechanical model, which is by the way already coded in a software realized by the same Authors. If efficiency and quickness reasons are adduced, then they must be proved.

In order to publish their manuscript in Royal Society Open Science, the Authors are recommended to revise the paper in order to clarify its main point. So far, the paper appears as a mere exercise of data manipulation.

Decision letter (RSOS-210568.R0)

Dear Dr Mou

The Editors assigned to your paper RSOS-210568 "Prediction model for aerodynamic coefficients of iced quad bundle conductors based on machine learning method" have now received comments from reviewers and would like you to revise the paper in accordance with the reviewer comments and any comments from the Editors. Please note this decision does not guarantee eventual acceptance.

Please submit your revised manuscript and required files (see below) no later than 21 days from today's (ie 20-Jul-2021) date. Note: the ScholarOne system will 'lock' if submission of the revision is attempted 21 or more days after the deadline. If you do not think you will be able to meet this deadline please contact the editorial office immediately.

on behalf of R. Kerry Rowe (Subject Editor)
openscience@royalsociety.org

Associate Editor Comments to Author:

Comments to the Author:

The reviewer comments suggest that there may be some merit in your work, but they each offer a number of suggestions for improvements. A particular point to note is that of the second reviewer who notes that it's not entirely clear what the purpose of the work is - given this, you should take care to revise the paper to clarify the scientific 'story' you're trying to tell: a reader should be able to understand the logic and flow of the work to gain the greatest value from your research. Please ensure that you provide a tracked-changes revision and point-by-point response to the reviewers in your revision.

Reviewer comments to Author:

Reviewer: 1

Comments to the Author(s)

It is interesting to establish a prediction model for aerodynamic coefficients of iced quad bundle conductors. The authors created datasets by using the Latin hypercube sampling (LHS) and numerical simulation. The extra-trees algorithm was also adopted to realize the mapping from the number of sub-conductors, wind attack angle, bundle spacing, ice accretion angle, ice thickness, wind velocity, and diameter of conductor to the aerodynamic coefficients of iced quad bundle conductors. The overall idea is good, but there are still some issues that need attention, as listed below:

1. In figure 6, it seems that the results of numerical simulation are not in good agreement with the wind tunnel test results. The lift coefficient curves of the two ways do not overlap when the wind attack angle is less than 50 degrees greater than 310 degrees. Previous studies have shown that the lift coefficient of the crescent-shaped iced conductors changes dramatically in the above-mentioned range of wind attack angle, so it needs to be treated with extreme caution. Such simulation results are used as training data, making the accuracy of the prediction results questionable. It is suggested to validate the in-house simulation software

against the commercial one, i.e., Fluent, before using the simulation results to train a machine learning model.

2. Please compare the computational cost of the numerical simulation and the prediction model. Since the aerodynamic coefficients of iced conductors could be well simulated by the in-house CFD software, it reduces the importance of developing the machine learning based prediction model. Please discuss and clarify this issue.

Reviewer: 2

Comments to the Author(s)

Review of the manuscript RSOS-210568, "Prediction model for aerodynamic coefficients of iced quad bundle conductors based on machine learning method" by Zheyue Mou, Bo Yan, Hanxu Yang, Daoda Cai, Guizao Huang, submitted for publication in Royal Society Open Science.

The paper proposes a prediction model based on machine learning to evaluate the aerodynamic coefficients of a quad bundle of iced conductors. Description of the numerical model which is used to build the dataset is first given. Then the prediction model, based on the work proposed by Geurts et al. [22], is realized and tested. Comparison of the galloping features of conductors, as evaluated using coefficients which are outcomes of both numerical and prediction procedures, is done.

The paper shows two distinct souls: on the one hand the first piece, which belongs to the field of mechanics, is interesting but it is a summary of other jobs already published by some of the Authors [12]; on the other hand, the second piece (Section 3), is the actual core of the paper and describes the machine learning algorithm. It belongs to the information technology field and, in the context of the problem, appears as pointless. In particular, it is not clear (or not clearly described) the necessity of creating such a machine learning structure to (not exactly) reproduce outcomes which are available by a mechanical model, which is by the way already coded in a software realized by the same Authors. If efficiency and quickness reasons are adduced, then they must be proved.

In order to publish their manuscript in Royal Society Open Science, the Authors are recommended to revise the paper in order to clarify its main point. So far, the paper appears as a mere exercise of data manipulation.

===PREPARING YOUR MANUSCRIPT===

===PREPARING YOUR REVISION IN SCHOLARONE===

- Ensure that your data access statement meets the requirements at <https://royalsociety.org/journals/authors/author-guidelines/#data>. You should ensure that you cite the dataset in your reference list. If you have deposited data etc in the Dryad repository, please include both the 'For publication' link and 'For review' link at this stage.
- If you are requesting an article processing charge waiver, you must select the relevant waiver option (if requesting a discretionary waiver, the form should have been uploaded at Step 3 'File upload' above).
- If you have uploaded ESM files, please ensure you follow the guidance at <https://royalsociety.org/journals/authors/author-guidelines/#supplementary-material> to include a suitable title and informative caption. An example of appropriate titling and captioning may be found at https://figshare.com/articles/Table_S2_from_Is_there_a_trade-off_between_peak_performance_and_performance_breadth_across_temperatures_for_aerobic_scope_in_teleost_fishes_/3843624.

Author's Response to Decision Letter for (RSOS-210568.R0)

See Appendix A.

RSOS-210568.R1 (Revision)

Review form: Reviewer 1

Is the manuscript scientifically sound in its present form?

Yes

Are the interpretations and conclusions justified by the results?

Yes

Is the language acceptable?

Yes

Do you have any ethical concerns with this paper?

No

Have you any concerns about statistical analyses in this paper?

No

Recommendation?

Accept with minor revision (please list in comments)

Comments to the Author(s)

The authors only calculated the gallop when the angle of attack is 45° . However, according to literature 12, the Den Hartog coefficient at this angle of attack is close to zero, which is not representative. Please compare the Den Hartog coefficients and the Nigol coefficients obtained from the numerical simulation and the test results under the conditions given in Figure 6, and provide proof that the deviation of gallop calculation caused by the two methods is acceptable when the wind attack angle interval is $160^\circ\sim 190^\circ$, $265^\circ\sim 280^\circ$, $310^\circ\sim 350^\circ$.

Review form: Reviewer 2

Is the manuscript scientifically sound in its present form?

Yes

Are the interpretations and conclusions justified by the results?

Yes

Is the language acceptable?

Yes

Do you have any ethical concerns with this paper?

No

Have you any concerns about statistical analyses in this paper?

No

Recommendation?

Accept as is

Comments to the Author(s)

The revised version of the manuscript can be considered for publication in RSOS as it is.

Decision letter (RSOS-210568.R1)

Dear Dr Mou

The Editors assigned to your paper RSOS-210568.R1 "Prediction model for aerodynamic coefficients of iced quad bundle conductors based on machine learning method" have now received comments from reviewers and would like you to revise the paper in accordance with the reviewer comments and any comments from the Editors. Please note this decision does not guarantee eventual acceptance.

Please submit your revised manuscript and required files (see below) no later than 21 days from today's (ie 24-Aug-2021) date. Note: the ScholarOne system will 'lock' if submission of the revision is attempted 21 or more days after the deadline. If you do not think you will be able to meet this deadline please contact the editorial office immediately.

on behalf of Prof R. Kerry Rowe (Subject Editor)
openscience@royalsociety.org

Associate Editor Comments to Author:

One of the reviewers agrees that you have fully responded to their concerns; however, the second reviewer continues to raise concerns regarding the work and has indicated they do not consider your response to their earlier comments to be sufficient to allow them to recommend the work. Given this, we'd like you to revise the paper to address their commentary - please be aware that it is not common for authors to be granted a second chance to revise the paper like this: you will not be given a further opportunity. Good luck with the revision and we'll look forward to receiving this in the near future.

Reviewer comments to Author:

Reviewer: 2

Comments to the Author(s)

The revised version of the manuscript can be considered for publication in RSOS as it is.

Reviewer: 1

Comments to the Author(s)

The authors only calculated the gallop when the angle of attack is 45°. However, according to literature 12, the Den Hartog coefficient at this angle of attack is close to zero, which is not

representative. Please compare the Den Hartog coefficients and the Nigol coefficients obtained from the numerical simulation and the test results under the conditions given in Figure 6, and provide proof that the deviation of gallop calculation caused by the two methods is acceptable when the wind attack angle interval is $160^{\circ}\sim 190^{\circ}$, $265^{\circ}\sim 280^{\circ}$, $310^{\circ}\sim 350^{\circ}$.

===PREPARING YOUR MANUSCRIPT===

===PREPARING YOUR REVISION IN SCHOLARONE===

Author's Response to Decision Letter for (RSOS-210568.R1)

See Appendix B.

RSOS-210568.R2 (Revision)

Review form: Reviewer 2

Is the manuscript scientifically sound in its present form?

Yes

Are the interpretations and conclusions justified by the results?

Yes

Is the language acceptable?

Yes

Do you have any ethical concerns with this paper?

No

Have you any concerns about statistical analyses in this paper?

No

Recommendation?

Accept as is

Comments to the Author(s)

The revised paper can be published as it is

Decision letter (RSOS-210568.R2)

Dear Dr Mou,

It is a pleasure to accept your manuscript entitled "Prediction model for aerodynamic coefficients of iced quad bundle conductors based on machine learning method" in its current form for publication in Royal Society Open Science. The comments of the reviewer(s) who reviewed your manuscript are included at the foot of this letter.

You can expect to receive a proof of your article in the near future. Please contact the editorial office (openscience@royalsociety.org) and the production office (openscience_proofs@royalsociety.org) to let us know if you are likely to be away from e-mail

contact -- if you are going to be away, please nominate a co-author (if available) to manage the proofing process, and ensure they are copied into your email to the journal.

on behalf of Prof R. Kerry Rowe (Subject Editor)
openscience@royalsociety.org

Reviewer comments to Author:
Reviewer: 2
Comments to the Author(s)
The revised paper can be published as it is

Appendix A

Response to associate editor's comments:

The reviewer comments suggest that there may be some merit in your work, but they each offer a number of suggestions for improvements. A particular point to note is that of the second reviewer who notes that it's not entirely clear what the purpose of the work is - given this, you should take care to revise the paper to clarify the scientific 'story' you're trying to tell: a reader should be able to understand the logic and flow of the work to gain the greatest value from your research. Please ensure that you provide a tracked-changes revision and point-by-point response to the reviewers in your revision.

Response: The authors thank the editor very much for his/her good suggestions. We have revised the manuscript carefully to highlight the purpose and significance of the work, and make the readers understand the logic and flow of the work more easily. The responses to the reviewers' comments are listed as the follows one by one. In addition, we read the guide and additional requirements the editor provided, and follow the rules of the journal carefully.

Response to reviewers' comments:

The authors thank the reviewers very much for their careful review and good suggestions, and have tried to revise the manuscript carefully. The responses to the comments are listed as the follows one by one.

Responses to comments of reviewer 1

It is interesting to establish a prediction model for aerodynamic coefficients of iced quad bundle conductors. The authors created datasets by using the Latin hypercube sampling (LHS) and numerical simulation. The extra-trees algorithm was also adopted to realize the mapping from the number of sub-conductors, wind attack angle, bundle spacing, ice accretion angle, ice thickness, wind velocity, and diameter of conductor to the aerodynamic coefficients of iced quad bundle conductors. The overall idea is good, but there are still some issues that need attention, as listed below:

(1) In figure 6, it seems that the results of numerical simulation are not in good agreement with the wind tunnel test results. The lift coefficient curves of the two ways do not overlap when the wind attack angle is less than 50 degrees greater than 310 degrees. Previous studies have shown that the lift coefficient of the crescent-shaped iced conductors changes dramatically in the above mentioned range of wind attack angle, so it needs to be treated with extreme caution. Such simulation results are used as training data, making the accuracy of the prediction results questionable. It is suggested to validate the in-house simulation software against the commercial one, i.e., Fluent, before using the simulation results to train a machine learning model.

Response: Thank the reviewer for his/her careful reviewing. Firstly, the difference between the coefficients obtained by the wind tunnel test and numerical simulation is inevitable. As indicated in the last paragraph of Section 2.3, “The differences between the aerodynamic coefficients obtained by the two methods may be induced by the simplification of stranded structure of conductor in the numerical model, and the stability of inflow wind field and measurement errors in the wind tunnel tests”.

Secondly, although the difference between the data obtained by the two methods exists, the data by the numerical simulation are acceptable for the investigation of galloping features of iced transmission lines as indicated in Ref. [12]. To clarify this point, the last sentence of Section 2.3 is modified as “However, the galloping of transmission lines depends on the changing laws of the aerodynamic coefficients, and the Den Hartog coefficients and Nigol coefficients [2] determined by the aerodynamic coefficients. Our previous research [12] demonstrates that the galloping features of iced transmission lines obtained by using the tested coefficients and simulated coefficients respectively are similar, especially the galloping mode and frequency, which can be used to guide the design of anti-galloping technology and devices. Therefore, the deviation is acceptable and the simulated coefficients can be used to investigate the galloping.”

Thirdly, the software is actually a secondarily developed software to quickly create the iced bundle conductor models and output the curves of the aerodynamic coefficients varying with wind attack angle, and the Fluent is called as solver by the software. To

clarify this point, the second paragraph of Section 2.1 is modified as “Recently the group of the authors secondarily developed a simulation software for aerodynamic characteristics of iced conductors, which calls the commercial computational fluid dynamics software Fluent as solver to simulate the air flow around the iced conductor, as shown in figure 2. With the secondarily developed software, the geometrical models of iced conductors with any shaped ice can be constructed quickly and the curves of the three aerodynamic coefficients varying with wind attack angle can be output automatically.”

(2) Please compare the computational cost of the numerical simulation and the prediction model. Since the aerodynamic coefficients of iced conductors could be well simulated by the in-house CFD software, it reduces the importance of developing the machine learning based prediction model. Please discuss and clarify this issue.

Response: In order to compare the computational cost of the numerical simulation and the prediction model, the sentence ‘It is noted that the average simulation time for obtaining the curves of the aerodynamic coefficients of an iced quad bundle conductor varying with wind attack angle is more than 30 hours using computer ThinkCentre M8600t with Intel(R) Core i7-6700.’ is added at the end of Section 2.1.

In addition, the sentence ‘Moreover, the average time for predicting the curves of the aerodynamic coefficients of an iced quad bundle conductor varying with wind attack angle by means of the trained model is only about 10 seconds, which is very quick compared with the average 30 hours by numerical simulation.’ is added at the end of Section 3.3.

The efficiency and quickness of the prediction model is further highlighted and thank the reviewer again for his/her good suggestions.

Responses to comments of reviewer 2

The paper proposes a prediction model based on machine learning to evaluate the aerodynamic coefficients of a quad bundle of iced conductors. Description of the

numerical model which is used to build the dataset is first given. Then the prediction model, based on the work proposed by Geurts et al. [22], is realized and tested. Comparison of the galloping features of conductors, as evaluated using coefficients which are outcomes of both numerical and prediction procedures, is done.

The paper shows two distinct souls: on the one hand the first piece, which belongs to the field of mechanics, is interesting but it is a summary of other jobs already published by some of the Authors [12]; on the other hand, the second piece (Section 3), is the actual core of the paper and describes the machine learning algorithm. It belongs to the information technology field and, in the context of the problem, appears as pointless. In particular, it is not clear (or not clearly described) the necessity of creating such a machine learning structure to (not exactly) reproduce outcomes which are available by a mechanical model, which is by the way already coded in a software realized by the same Authors. If efficiency and quickness reasons are adduced, then they must be proved.

In order to publish their manuscript in Royal Society Open Science, the Authors are recommended to revise the paper in order to clarify its main point. So far, the paper appears as a mere exercise of data manipulation.

Response: Thank the reviewer for his/her careful reviewing. The aim of this paper is to create a prediction model for aerodynamic coefficients of iced conductors by means of the machine learning method, which can reduce time and cost effectively.

To clarify the necessity of creating the prediction model (machine learning structure), several modifications are made in the Abstract and the Introduction: (1) The sentence “The developed efficient prediction model for the aerodynamic coefficients of iced quad bundle conductors plays an important role in the quick investigation, prediction and early warning of galloping.” is added at the end of the Abstract. (2) Two sentences are inserted into the first paragraph of the Introduction: “Quick investigation and prediction of galloping features of transmission lines are very important for the development of anti-galloping technique and early warning system.” and “Therefore, the creation of a prediction model to quickly determine the aerodynamic coefficients of iced conductors under different parameters is urgent for the investigation and prediction

of galloping features as well as the development of anti-galloping technique and early warning system.”

To demonstrate the efficiency and quickness, (1) the sentence ‘It is noted that the average simulation time for obtaining the curves of the aerodynamic coefficients of an iced quad bundle conductor varying with wind attack angle is more than 30 hours using computer ThinkCentre M8600t with Intel(R) Core i7-6700.’ is added at the end of Section 2.1, (2) the sentence ‘Moreover, the average time for predicting the curves of the aerodynamic coefficients of an iced quad bundle conductor varying with wind attack angle by means of the trained model is only about 10 seconds, which is very quick compared with the average 30 hours by numerical simulation.’ is added at the end of Section 3.3. (3) The accuracy of the model is also discussed in the last paragraph of Section 3.3.

The authors have tried to revise the paper carefully and highlight the main point and significance of the work, and thank the reviewer again for his/her good suggestions.

Appendix B

Response to associate editor's comments:

One of the reviewers agrees that you have fully responded to their concerns; however, the second reviewer continues to raise concerns regarding the work and has indicated they do not consider your response to their earlier comments to be sufficient to allow them to recommend the work. Given this, we'd like you to revise the paper to address their commentary - please be aware that it is not common for authors to be granted a second chance to revise the paper like this: you will not be given a further opportunity. Good luck with the revision and we'll look forward to receiving this in the near future.

Response: The authors thank the editor very much for his/her work. We have revised the manuscript carefully and tried to address the reviewers' comments fully. The responses to the comments are listed as the follows one by one.

Responses to comments of reviewer 1

The authors only calculated the gallop when the angle of attack is 45° . However, according to literature [12], the Den Hartog coefficient at this angle of attack is close to zero, which is not representative. Please compare the Den Hartog coefficients and the Nigol coefficients obtained from the numerical simulation and the test results under the conditions given in Figure 6, and provide proof that the deviation of gallop calculation caused by the two methods is acceptable when the wind attack angle interval is $160^\circ\sim 190^\circ$, $265^\circ\sim 280^\circ$, $310^\circ\sim 350^\circ$.

Response: Thank the reviewer for his/her careful reviewing and good suggestions. According to [12], "the Den Hartog coefficients of sub-conductors 3 and 4 are negative in the angle range of $40^\circ\sim 60^\circ$ ". The galloping of iced quad bundle conductor line is induced when the initial wind attack angle is 45° .

As indicated in the first paragraph of Section 2.4, the aerodynamic coefficients of the four sub-conductors of iced quad bundle conductor are obviously different due to the wake influence, and the Den Hartog coefficients and the Nigol coefficients are different too, which have important influence on the galloping of bundle conductor lines

[6]. To simulate the galloping of an iced quad-bundle conductor line, we need to investigate the Den Hartog coefficients and the Nigol coefficients of the four sub-conductors. So these two coefficients of the four sub-conductor with 12mm-thickness ice against the wind attack angle based on the numerical simulation and wind tunnel test are compared in figure 7. As the suggestion of the reviewer, the Den Hartog coefficient of all or some sub-conductors are negative in the angle ranges of $160^{\circ}\sim 190^{\circ}$, $265^{\circ}\sim 280^{\circ}$ and $310^{\circ}\sim 350^{\circ}$. In order to prove the acceptance of aerodynamic coefficients by numerical simulation further, the aerodynamic coefficients by the simulation and test are applied to simulate the galloping of a typical transmission line with the initial attack angle in the ranges of $160^{\circ}\sim 190^{\circ}$, $265^{\circ}\sim 280^{\circ}$ and $310^{\circ}\sim 350^{\circ}$. For the limitation of paper length, the Den Hartog coefficients and the Nigol coefficients of the four sub-conductors with other ice thickness under the conditions given in Figure 6 are not given out.

The detailed analysis process and simulated results are added in the paragraphs 2~5 of Section 2.3. According to the reviewer's suggestion, the authors revised Section 2.3 greatly to provide proof that the deviation of the galloping simulated based on the aerodynamic coefficients by the two methods is acceptable when the initial wind attack angles are in the ranges of $160^{\circ}\sim 190^{\circ}$, $265^{\circ}\sim 280^{\circ}$ and $310^{\circ}\sim 350^{\circ}$.

In addition, the numberings of the figures after figure 6 and the formulas after formula (1) are changed.

Responses to comments of reviewer 2

Comments to the Author(s): The revised version of the manuscript can be considered for publication in RSOS as it is.

Response: Thank the reviewer very much for his/her careful reviewing and recommended publication of the manuscript.